# Management of XEN Gel Stent Exposure with Conjunctival Erosion via Rotational Conjunctival Flap and Amniotic Membrane Transplantation—A Case Report

**DOI:** 10.3390/medicina58111581

**Published:** 2022-11-02

**Authors:** Chang Kyu Lee, Je Hyun Seo, Su-Ho Lim

**Affiliations:** 1Department of Ophthalmology, Ulsan University Hospital, University of Ulsan College of Medicine, Ulsan 44033, Korea; 2Veterans Medical Research Institute, Veterans Health Service Medical Center, Seoul 05368, Korea; 3Department of Ophthalmology, Daegu Veterans Health Service Medical Center, Daegu 42835, Korea

**Keywords:** XEN gel stent, conjunctival erosion, amniotic membrane transplantation, rotational conjunctival flap, anterior segment OCT

## Abstract

*Background*: Despite its proven effectiveness and safety profile, the XEN gel stent (Allergan Inc., CA, USA) for minimally invasive glaucoma surgery (MIGS) has a probability of postoperative complications, including postoperative hypotony, hyphema, stent migration, stent obstruction, bleb fibrosis, and fibrin formation. In particular, the use of adjunctive Mitomycin-C (MMC) might be associated with bleb-related complications, including conjunctival erosion, XEN gel stent exposure, and blebitis. However, there are few studies on XEN gel stent exposure and its management. We describe a case of XEN gel stent exposure with conjunctival erosion 18 months postoperatively, which resolved effectively after combination treatment with a rotational conjunctival flap and amniotic membrane transplantation. *Case presentation*: A 74-year-old Korean male patient with diabetes and hypertension underwent uncomplicated ab interno XEN gel stent implantation with a subconjunctival injection of 0.1 cc of 0.02% MMC and presented with low intraocular pressure (IOP) with a well-functioning filtering bleb. Periocular pain and tearing developed 18 months after the initial operation, with mild deterioration of visual acuity to 20/100. Despite conservative medical treatment, the conjunctival erosion was not relieved. Anterior segment optical coherence tomography (AS-OCT) revealed an exposed XEN gel stent with conjunctival erosion. We performed bleb revision surgery using a rotational conjunctival flap and amniotic membrane transplantation. Slit-lamp examination and AS-OCT showed a well-formed moderate bleb without leakage, and IOP continued to be well controlled (14 mm Hg with latanoprost) until six months after bleb revision. *Conclusions*: This case report highlights the importance of careful examination, including slit-lamp examination, the Seidel test, and AS-OCT, to identify accurate anatomical positioning and to monitor ocular surface changes after XEN gel stent implantation with MMC or 5-FU. Combination treatment (rotational conjunctival flap and amniotic membrane transplantation) may be relatively safe for persistent XEN gel stent exposure.

## 1. Introduction

Minimally invasive glaucoma surgery (MIGS) has changed the surgical treatment paradigm for glaucoma. The XEN45 (Allergan, CA, USA) gel stent is a 6 mm-long hydrophilic tube made of collagen-derived gelatin cross-linked with glutaraldehyde [1,2,3,4]. XEN implantation may often be considered an alternative to traditional glaucoma surgeries, with a lower risk of significant complications such as hypotony and hemorrhage [3,4]. The adjunctive subconjunctival mitomycin-C (MMC) is widely used because XEN implants have small lumens and the likelihood of postoperative conjunctival scarring, which requires medical treatment, more frequent needling revision, and bleb revision [3,4]. As a result, the use of adjunctive Mitomycin-C (MMC) might be associated with bleb-related complications, including conjunctival erosion, XEN gel stent exposure, blebitis, and surgically induced necrotizing scleritis (SINS) [5,6,7,8]. To date, few studies have explored XEN gel stent exposure and its management [5,6,9]. Herein, we describe the first case of XEN gel stent exposure with conjunctival erosion 18 months postoperatively, which resolved effectively after combination treatment with a rotational conjunctival flap and amniotic membrane transplantation.

## 2. Case Presentation

The study adhered to the principles of the Declaration of Helsinki. Written informed consent for publication of the report and photographs was obtained from the patient. This case study was approved by the Institutional Review Board of the Daegu Veterans Health Service Medical Center.

### 2.1. Initial Presentation

A 74-year-old Korean male patient with diabetes and hypertension was referred to the Daegu Veterans Health Service Medical Center for borderline intraocular pressure (IOP) control and progressive loss of vision in his right eye. The patient was diagnosed with primary open-angle glaucoma (POAG) with proliferative diabetic retinopathy (PDR). He had been receiving maximal medical treatment for three years, including a combination of brinzolamide and timolol (Elazop^®^, *Novartis* Pharmaceuticals, USA), brimonidine tartrate 0.15% (Alphagan^®^, Allergan Inc., USA), and tafluprost (Taflotan^®^, Santen Pharmaceuticals, JAPAN). At the time of presentation, his best-corrected visual acuity was 0.4 logMAR (20/50 Snellen), and intraocular pressure (IOP) was 24 mm Hg upon Goldmann applanation tonometry in the right eye. Slit-lamp examination revealed a deep and quiet anterior chamber and well-positioned intraocular lens. The cup-to-disc ratio was 0.9, with superior and inferior notching and scattered laser scars due to proliferative diabetic retinopathy in the right eye. The Humphrey visual field test showed glaucomatous damage (Figure 1; mean deviation: −17.95 dB, pattern deviation: 10.44 dB, visual field index: 54%). Spectral-domain optical coherence tomography also demonstrated generally reduced RNFL thickness. Gonioscopic examination revealed a wide iridocorneal angle without neovascularization (D40r; Spaeth classification).

### 2.2. The Surgical Procedure for Initial XEN Gel Stent Implantation

The XEN gel stent was implanted superonasally ab interno, via the usual manner [1,2,3,4]. Briefly, under topical anesthesia, 0.1 mL MMC (0.2 mg/mL) was injected using a 30-gauge needle in the superonasal quadrant. The intended area of placement in the superonasal quadrant, 3 mm from the limbus, was marked. Then, an ab interno XEN gel stent was implanted using a mirrored gonioscope (1 mm in the anterior chamber, and 2.0 mm tunneled through the sclera). The ophthalmic viscoelastic device was removed from the anterior chamber. Finally, we checked bleb morphology and function via the forced infusion of a balanced salt solution. At the end of the surgery, we found a mild nasally located XEN gel implant due to cyclotorsion. The patient was started on levofloxacin and prednisolone acetate four times daily the next day.

### 2.3. XEN Gel Stent Exposure with Conjunctival Erosion

The IOP was 11 mm Hg on day 1 and 10 mm Hg on day 7. Slit-lamp examination and anterior segment optical coherence tomography (AS-OCT) revealed a well-functioning bleb (Figure 2).

During the follow-up period, the hypovascular bleb gradually changed into an avascular bleb. Periocular pain and tearing developed 18 months after the initial operation, with mild deterioration of visual acuity from 20/50 (LogMAR 0.4) to 20/100 (LogMAR 0.7). Slit-lamp examination revealed a small defect in the conjunctiva, corresponding to the exit point of the XEN. The Seidel test result was weakly positive; however, the patient’s IOP remained acceptable at 15 mm Hg and there was no evidence of blebitis (Figure 3).

## 3. Management and Outcomes

### 3.1. Management for XEN Gel Stent Exposure with Conjunctival Erosion

Topical antibiotics, lubricants, and autologous serum eye drops were initially administered to facilitate wound healing. In detail, 20 mL of blood was obtained from the antebrachial vein and centrifuged for 20 min at 1500 rpm, and the separated serum was diluted with normal saline to a 20% concentration. All topical eye drops were applied to the eye with a 5 min interval four times a day. However, these medical treatments did not improve the conjunctival defect or ocular surface. Thus, we decided to perform a surgical bleb revision for XEN gel stent exposure. The management comprised three aspects: (1) the prevention of wound leakage (conjunctival flap), (2) the facilitation of wound healing (use of the amniotic membrane), and (3) minimal manipulation of the XEN gel stent itself, considering the fragile property of hydrated collagen implants. The plan is summarized in Figure 4.

Under a surgical microscope, the conjunctival defect was identified, the amount of bare sclera was measured using a caliper, and the corresponding conjunctival graft was rotated from the inferonasal conjunctiva. The rotational conjunctival flap was first anchored using interrupted 10-0 nylon sutures and covered using a complementary amniotic membrane graft with fibrin glue (epithelial side up). Finally, oversized temporary amniotic membrane transplantation (epithelial side down) was performed over the entire avascular filtering bleb using a 10-0 nylon continuous locking suture to minimize additional conjunctival or bleb injuries during the procedure.

### 3.2. Surgical Outcomes

Three months following bleb revision surgery with conjunctival flap and amniotic membrane transplantation in the right eye, IOP was 15 mm Hg with preservative-free latanoprost (Monoprost^®^, Samil Pharmaceutical Co., Ltd., Seoul, Korea) once daily. Slit-lamp examination and AS-OCT revealed a well-formed moderate bleb with no leakage (Figure 5). Additionally, visual field was stable without progression at 6 months postoperatively.

## 4. Discussion

The XEN45 gel stent has been recognized as a safe and effective micro-invasive glaucoma surgery procedure, even in advanced and refractory glaucoma [1,2,3,4]. Intraoperative adjunctive MMC and postoperative needling revision using anti-fibrotic agents are widely used in XEN gel stent implantation for wound modulation and the prevention of bleb fibrosis [1,2,3,4]. These procedures might be associated with the possibility of postoperative bleb-related complications, including over-filtering blebs, blebitis, bleb dysesthesia, bleb leakage, and endophthalmitis [3,4,5,6,7,8,9]. However, reports on XEN exposure and its management are limited [5,6,9].

In our patient, XEN gel stent exposure and conjunctival erosion were observed at 18 months postoperatively. The detection timing for stent exposure or extrusion varied from 1 month to 14 months in previous case reports. Arnould et al. reported recurrent XEN exposure three months postoperatively (two months after needling revision) [5]. Lapira al. reported extrusion and breakage of an XEN stent with endophthalmitis 3.5 months postoperatively [6]. Similarly, Karri et al. reported endophthalmitis following XEN stent exposure postoperatively after 4 months [7]. Kingston et al. reported that infective necrotizing scleritis developed 14 months after the initial surgery [8].

Possible mechanisms for the initial conjunctival erosions are as follows: (1) the use of the anti-metabolite MMC; (2) the ab interno approach; (3) the subconjunctival position; (4) the long-term use of topical anti-glaucoma medications; and (5) mechanical stress, such as the elderly patient rubbing the eye with their hands. First, the use of anti-metabolites enhances the success rate of filtering surgery by reducing the wound-healing process. However, the use of anti-metabolites might also increase the risk of bleb-related complications, such as a thin-walled cystic bleb, or surgically induced necrotizing scleritis [4,5,6,8]. In this patient, we used a 0.1 mL subconjunctival injection of 0.02% MMC. Second, the ab interno approach may also contribute to conjunctival erosions [4,5,6,7]. In XEN gel stent implantation with an ab externo approach and conventional trabeculectomy, MMC is usually applied using MMC-soaked cellulose sponges and wash-out. However, these procedures are not applicable with an ab interno approach, because the MMC is injected directly into the subconjunctival space and remains in this space at the end of surgery without washout [3,4,6]. This difference in the approach may cause more vulnerable and avascular bleb formation. Third, we considered the positioning of the XEN subconjunctival space versus sub-Tenon’s space [3]. In this case, the XEN gel implant was placed in the subconjunctival space. Compared to sub-Tenon’s position, a subconjunctival placement may have a higher risk of micro-trauma and subsequent infection, whereas sub-Tenon’s placement has a higher chance of the tip of the stent becoming entangled in Tenon’s position and subsequent bleb failure [4,7]. Long-term use of topical drugs can induce changes in the conjunctiva and ocular surface, including conjunctival thinning and metaplasia. These qualitative changes and conjunctival thinning may also contribute to conjunctival erosion.

In addition to pre-existing risk factors that might contribute to the formation of avascular thin blebs, mechanical stress could have aggravated conjunctival injury due to foreign-body sensation in this elderly patient [6] with degenerative conjunctival changes, such as Meibomian gland dysfunction, decreased tear film volume, tear film instability, and coexisting ocular disease with diabetes and hypertension. Based on AS-OCT examination, we hypothesized that all these mechanisms might induce XEN gel stent exposure with conjunctival erosion. Similarly, Arnould et al. suggested that morphological changes in OCT after XEN gel stent implantation may help guide bleb revision and the management of conjunctival erosion and device extrusion [5]. In our case, AS-OCT examination demonstrated XEN gel stent exposure without conjunctival coverage and surgical results. In this context, the proper use of AS-OCT for evaluating and monitoring bleb morphology could provide additional information regarding XEN gel stent positioning, bleb morphology, and the status of the conjunctiva or Tenon’s capsule.

Various conservative treatment options, such as pressure patching, contact lenses, topical lubricants, autoserums, and ointments, have been considered to resolve these problems [4,5,8,9]. However, in our patient, conservative treatment was unsuccessful. Therefore, we decided to perform surgical bleb revision. Arnould et al. [5] reported excision of a scarred bleb, free conjunctival graft, XEN gel stent trimming by 1.5 mm and complementary amniotic graft. Kingston et al. reported removal of the XEN gel stent, Tutoplast plugging into the scleral opening, amniotic membrane graft over the ischemic scleral bed, and fixation to healthy conjunctival and limbal tissues. [8] Another study reported ab interno repositioning of the stent through the anterior chamber and direct suturing of the conjunctival defect [9].

We performed combination treatment (rotational conjunctival flap and amniotic membrane transplantation). This management comprised three aspects: (1) dealing with wound leakage, (2) the facilitation of wound healing, and (3) minimal manipulation of the XEN gel stent itself. First, a rotational conjunctival flap was used because the conjunctival local flap (advancement or rotational flap) has been shown to provide successful resolution of bleb-associated complications in conventional filtration surgery [4,10]. In our case, the conjunctiva surrounding the XEN exposure with conjunctival defect also revealed a thin avascular bleb. We consider that free grafts alone, without bleb resection, may have a higher risk of recurrence of bleb leak and structural weakness in another harvested quadrant [4,5]. The conjunctival advancement flap had the possibility of excessive traction force [4] and lower filtering bleb height in our patient. Second, the amniotic membrane could act as a biological bandage contact lens to decrease mechanical stress and promote wound healing via anti-inflammatory action, anti-scarring action, and various growth factors such as EGF, NGF, and IGF [11]. In our patient, the amniotic membrane was sutured to cover both healthy host tissue and the site of interest, including thin avascular blebs and a conjunctival flap overlying the conjunctival defect. Liu et al. suggested that an ideal procedure for repairing bleb leaks is to support fragile conjunctival tissue and suppress exaggerated bleb scarring [11]. In this regard, AM has exhibited promising results in repairing bleb leak regardless of the epithelial side being up or down, over or under the conjunctiva, or with or without bleb excision after trabeculectomy [11]. Third, we attempted to perform minimal manipulation of the XEN gel stent itself. The flexibility and properties of stents differ depending on collagen hydration. Moreover, the XEN gel stent is designated as the ratio of the length to the lumen diameter (6 mm to 45 µm) to regulate optimal outflow, considering the Hagen–Poiseuille law [1,2,3,4]. The direct manipulation of hydrated collagen stents might pose a risk of breakage or cutting [4,6]. Thus, we adopted a rotational conjunctival flap and amniotic membrane patch graft using fibrin glue to minimize suture-induced inflammation, and an oversized temporary complimentary amniotic membrane using 10-0 continuous nylon sutures.

This study has some limitations. First, the results of our case report should be confirmed in a larger series, and our case reports provide excellent ocular surface reconstruction and acceptable IOP control without bleb leakage. Second, long-term follow-up is needed to monitor the ocular surface, considering the possibility of recurrent leakage despite minimal manipulation of the XEN gel stent.

This case report highlights the importance of careful examination, including slit-lamp examination, the Seidel test, and AS-OCT, to identify accurate anatomical positioning and to monitor ocular surface changes after XEN gel stent implantation with MMC or 5-FU. We also demonstrate the potential complications of XEN gel stent exposure and its management in addition to previously reported surgical techniques [5,8,9]. Thus, ophthalmic surgeons should also consider the optimal concentration of MMC according to the approach methods and patient profile, such as age, race, and previous surgical history, which determine the anticipated scarring tendency [3].

## 5. Conclusions

To the best of our knowledge, this is the first case report on the efficacy of combination treatment (rotational conjunctival flap and amniotic membrane transplantation) without manipulation of the XEN gel stent itself for persistent XEN gel stent exposure with conjunctival erosion. This combination therapy may be a relatively safe rescue treatment for XEN gel stent exposure.

## Figures and Tables

**Figure 1 medicina-58-01581-f001:**
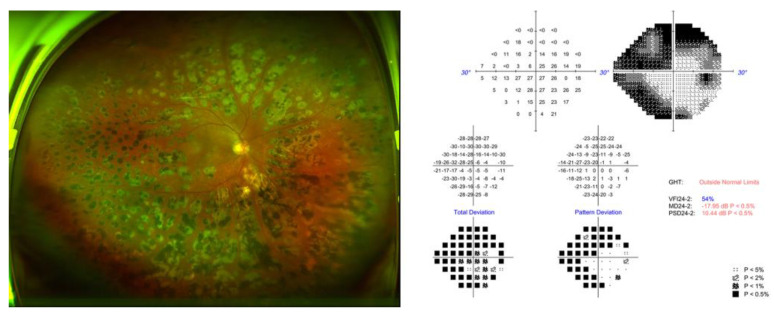
Preoperative ultra-wide-field fundus photograph and visual field. (GHT, glaucoma hemifield test; MD, mean deviation; PSD, pattern standard deviation).

**Figure 2 medicina-58-01581-f002:**
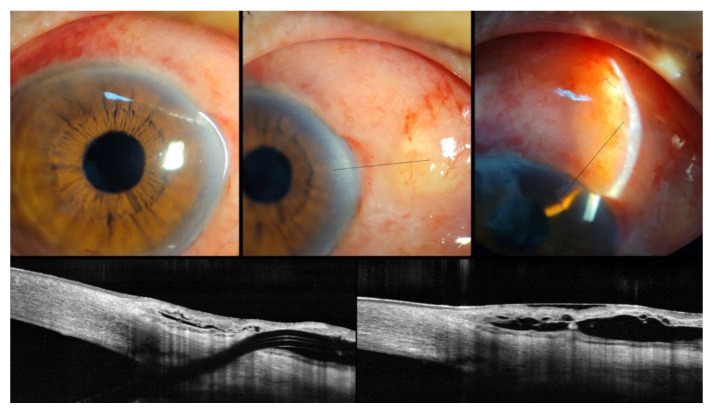
Anterior segment photography and AS-OCT demonstrated relatively well-functioning filtering bleb at 7 days after XEN implantation. (Black line: linear scan line on AS-OCT).

**Figure 3 medicina-58-01581-f003:**
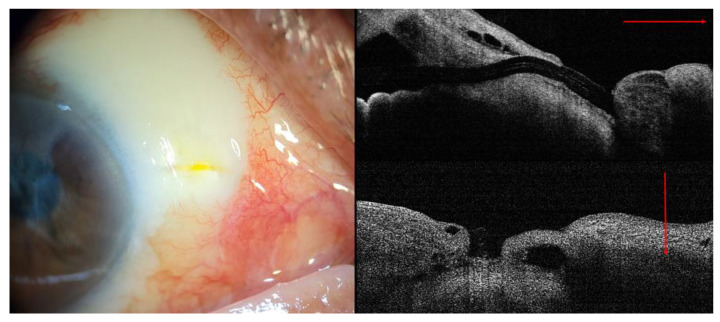
Slit-lamp examination and AS-OCT revealed the XEN gel stent exposure with small conjunctival defect due to conjunctival erosion (red line: linear scan on AS-OCT).

**Figure 4 medicina-58-01581-f004:**
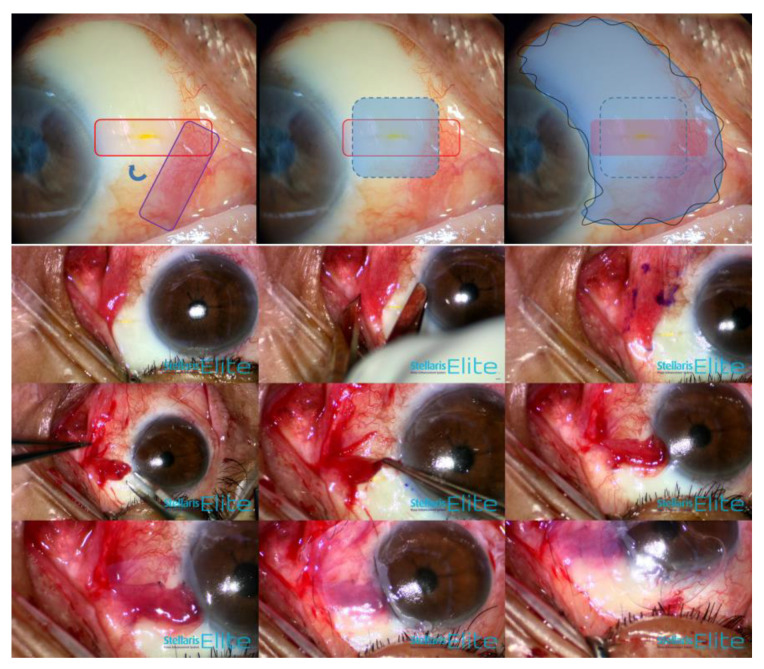
Schematic plan for rotational conjunctival flap and the amniotic membrane transplantation using fibrin glue and anchoring suture (purple: harvested donor conjunctiva; red: rotational conjunctival flap; light blue dotted line: complementary amniotic membrane graft using fibrin glue; blue line: temporary amniotic membrane transplantation using 10-0 nylon continuous locking suture (black line).

**Figure 5 medicina-58-01581-f005:**
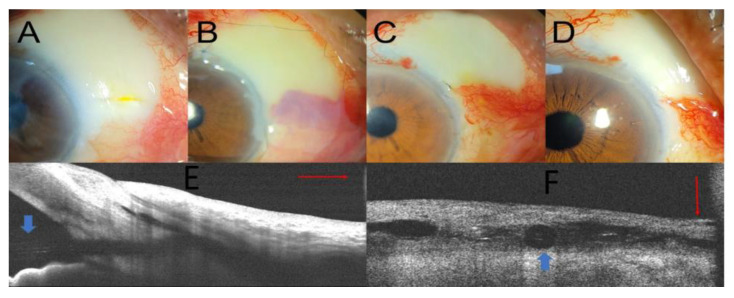
Preoperative anterior segment photography (**A**) and postoperative photographs on day 10 and at 1 month and 3 months (**B**–**D**). AS-OCT (**E**,**F**) demonstrated the well-covered conjunctiva over the XEN gel stent postoperatively after 1 month (blue arrow: XEN gel stent implant; red line: linear scan line on AS-OCT).

## Data Availability

All data generated or analyzed during this study are included in this article. The datasets used and/or analyzed during the current study are available from the corresponding author upon reasonable request.

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
