# Peer review of "Management of XEN Gel Stent Exposure with Conjunctival Erosion via Rotational Conjunctival Flap and Amniotic Membrane Transplantation—A Case Report"

_medicina, 2022, doi:10.3390/medicina58111581_

Round 1

Reviewer 1 Report

This is well written case presentation of xen gel implant exposure  solving. Literature related to this problem is still very scarce, so this work will be of importance to glaucomatologists.

The only complaint I have is that I personally would like to read in detail the use of auto serum in this case.

PAGE 4 LINE 111 – auto-serum - please describe in detail

Reviewer 2 Report

The study presents a case of conjunctival erosion following implantation of Xen device.

Line 64 et seq. reports that the patient is on maximal topical therapy for glaucoma. It is not specified how long this therapy has been in place.

This maximal therapy often causes qualitative changes in the conjunctiva and may result in thinning of the conjunctiva. In the discussion (line 159 et seq.), this clinical situation is not considered among the possible contributing factors in causing conjunctival erosion.

The patient is on maximal therapy for advanced glaucoma, however, after amniotic membrane transplantation surgery, the visual field situation is not shown
